# STREAMING PROBABILISTIC DEEP TENSOR FACTORIZATION

## ABSTRACT

Despite the success of existing tensor factorization methods, most of them conduct a multilinear decomposition, and rarely exploit powerful modeling frameworks, like deep neural networks, to capture a variety of complicated interactions in data. More important, for highly expressive, deep factorization, we lack an effective approach to handle streaming data, which are ubiquitous in real-world applications. To address these issues, we propose SPIDER, a Streaming ProbabilistIc Deep tEnsoR factorization method. We first use Bayesian neural networks (NNs) to construct a deep tensor factorization model. We assign a spike-and-slab prior over each NN weight to encourage sparsity and to prevent overfitting. We then use multivariate Delta's method and moment matching to approximate the posterior of the NN output and calculate the running model evidence, based on which we develop an efficient streaming posterior inference algorithm in the assumed-density-filtering and expectation propagation framework. Our algorithm provides responsive incremental updates for the posterior of the latent factors and NN weights upon receiving new tensor entries, and meanwhile select and inhibit redundant/useless weights. We show the advantages of our approach in four real-world applications.

## 1 Introduction

Tensor factorization is a fundamental tool for multiway data analysis. While many tensor factorization methods have been developed (Tucker, 1966; Harshman, 1970; Chu & Ghahramani, 2009; Kang et al., 2012; Choi & Vishwanathan, 2014), most of them conduct a mutilinear decomposition and are incapable of capturing complex, nonlinear relationships in data. Deep neural networks (NNs) are a class of very flexible and powerful modeling framework, known to be able to estimate all kinds of complicated (*e.g.,* highly nonlinear) mappings. The most recent work (Liu et al., 2018; 2019) have attempted to incorporate NNs into tensor factorization and shown a promotion of the performance, in spite of the risk of overfitting the tensor data that are typically sparse.

Nonetheless, one critical bottleneck for NN based factorization is the lack of effective approaches for streaming data. In practice, many applications produce huge volumes of data at a fast pace (Du et al., 2018). It is extremely costly to run the factorization from scratch every time when we receive a new set of entries. Some privacy-demanding applications (*e.g.,* SnapChat) even forbid us from revisiting the previously seen data. Hence, given new data, we need an effective way to update the model incrementally and promptly.

A general and popular approach is streaming variational Bayes (SVB) (Broderick et al., 2013), which integrates the current posterior with the new data, and then estimates a variational approximation as the updated posterior. Although SVB has been successfully used to develop the state-of-the-art multilinear streaming factorization (Du et al., 2018), it does not perform well for (deep) NN based factorization. Due to the nested linear and nonlinear coupling of the latent embeddings and NN weights, the variational model evidence lower bound (ELBO) that SVB maximizes is analytically intractable and we have to seek for stochastic optimization, which is unstable and hard to diagnose the convergence. Consequently, the posterior updates are often unreliable and inferior, and in turn hurt the subsequent updates, leading to poor model estimations finally.

To address these issues, we propose SPIDER, a streaming probabilistic deep tensor factorization method that not only exploits NN's expressive power to capture intricate relationships, but also provides efficient, high-quality posterior updates for streaming data. Specifically, we first use Bayesian

neural networks to build a deep tensor factorization model, where the input is the concatenation of the associated factors in each tensor entry and the NN output predicts the entry value. To reduce the risk of overfitting, we place a spike-and-slab prior over each NN weight to encourage sparsity. For streaming inference, we use multivariate Delta's method (Bickel & Doksum, 2015) that employs a first-order Taylor expansion of the NN output to analytically compute its moments, and match the moments to obtain the its current posterior and the running model evidence. We then use back-propagation to calculate the gradient of the log evidence, with which we match the moments and update the posterior of the embeddings and NN weights in the assumed-density-filtering (Boyen & Koller, 1998) framework. Finally, after processing all the newly received entries, we update the spike-and-slab prior approximation with expectation propagation (Minka, 2001a) to select and inhibit redundant/useless weights. In this way, the incremental posterior updates are deterministic, reliable and efficient.

For evaluation, we examined SPIDER on four real-world large-scale applications, including both binary and continuous tensors. We compared with the state-of-the-art streaming tensor factorization algorithm (Du et al., 2018) based on a multilinear form, and streaming nonlinear factorization methods implemented with SVB. In both running and final predictive performance, our method consistently outperforms the competing approaches, mostly by a large margin. The running accuracy of SPIDER is also much more stable and smooth than the SVB based methods.

## 2 Background

**Tensor Factorization**. We denote a $K$-mode tensor by $\mathcal{Y} \in \mathbb{R}^{d_1 \times \cdots \times d_K}$, where mode $k$ includes $d_k$ nodes. We index each entry by a tuple $\mathbf{i} = (i_1, \ldots, i_K)$, which stands for the interaction of the corresponding $K$ nodes. The value of entry $\mathbf{i}$ is denoted by $y_{\mathbf{i}}$. To factorize the tensor, we represent all the nodes by $K$ latent embedding matrices $\mathcal{U} = \{\mathbf{U}^1, \ldots, \mathbf{U}^K\}$, where each $\mathbf{U}^k = [\mathbf{u}_1^k, \ldots, \mathbf{u}_{d_k}^k]^\top$ is of size $d_k \times r_k$, and each $\mathbf{u}_j^k$ is the embedding vector of node $j$ in mode $k$. The goal is to use $\mathcal{U}$ to recover the observed entries in $\mathcal{Y}$. To this end, the classical Tucker factorization (Tucker, 1966) assumes $\mathcal{Y} = \mathcal{W} \times_1 \mathbf{U}^1 \times_2 \ldots \times_K \mathbf{U}^K$, where $\mathcal{W} \in \mathbb{R}^{r_1 \times \cdots \times r_K}$ is a parametric tenor and $\times_k$ the mode-$k$ tensor matrix multiplication (Kolda, 2006), which resembles the matrix-matrix multiplication. If we set all $r_k = r$ and $\mathcal{W}$ to be diagonal, Tucker factorization becomes CANDECOMP/PARAFAC (CP) factorization (Harshman, 1970). The element-wise form is $y_{\mathbf{i}} = \sum_{j=1}^r \prod_{k=1}^M u_{i_k,j}^k = (\mathbf{u}_{i_1}^1 \circ \ldots \circ \mathbf{u}_{i_M}^M)^\top \mathbf{1}$, where $\circ$ is the Hadamard (element-wise) product and $\mathbf{1}$ the vector filled with ones. We can estimate the embeddings $\mathcal{U}$ by minimizing a loss function, *e.g.,* the mean squared error in recovering the observed elements in $\mathcal{Y}$.

**Streaming Model Estimation**. A general and popular framework for incremental model estimation is streaming variational Bayes(SVB) (Broderick et al., 2013), which is grounded on the incremental version of Bayes' rule,

$$p(\boldsymbol{\theta}|\mathcal{D}_{\text{old}} \cup \mathcal{D}_{\text{new}}) \propto p(\boldsymbol{\theta}|\mathcal{D}_{\text{old}})p(\mathcal{D}_{\text{new}}|\boldsymbol{\theta}) \tag{1}$$

where $\boldsymbol{\theta}$ are the latent random variables in the probabilistic model we are interested in, $\mathcal{D}_{\text{old}}$ all the data that have been seen so far, and $\mathcal{D}_{\text{new}}$ the incoming data. SVB approximates the current posterior $p(\boldsymbol{\theta}|\mathcal{D}_{\text{old}})$ with a variational posterior $q_{\text{cur}}(\boldsymbol{\theta})$. When the new data arrives, SVB integrates $q_{\text{cur}}(\boldsymbol{\theta})$ with the likelihood of the new data to obtain an unnormalized, blending distribution,

$$\tilde{p}(\boldsymbol{\theta}) = q_{\text{cur}}(\boldsymbol{\theta})p(\mathcal{D}_{\text{new}}|\boldsymbol{\theta}) \tag{2}$$

which can be viewed as approximately proportional to the joint distribution $p(\boldsymbol{\theta}, \mathcal{D}_{\text{old}} \cup \mathcal{D}_{\text{new}})$. To conduct the incremental update, SVB uses $\tilde{p}(\boldsymbol{\theta})$ to construct a variational ELBO (Wainwright et al., 2008), $\mathcal{L}(q(\boldsymbol{\theta})) = \mathbb{E}_q[\log(\tilde{p}(\boldsymbol{\theta})/q(\boldsymbol{\theta}))]$, and maximizes the ELBO to obtain the updated posterior, $q^* = \operatorname{argmax}_q \mathcal{L}(q)$. This is equivalent to minimizing the Kullback-Leibler (KL) divergence between $q$ and the normalized $\tilde{p}(\boldsymbol{\theta})$. We then set $q_{\text{cur}} = q^*$ and prepare the update for the next batch of new data. At the beginning (when we do not receive any data), we set $q_{\text{cur}} = p(\boldsymbol{\theta})$, the original prior in the model. For efficiency and convenience, a factorized variational posterior $q(\boldsymbol{\theta}) = \prod_j q(\theta_j)$ is usually adopted to fulfill cyclic, closed-form updates. For example, the state-of-the-art streaming tensor factorization, POST (Du et al., 2018), uses the CP form to build a Bayesian model, and applies SVB to update the posterior of the embeddings incrementally when receiving new tensor entries.

## 3 Bayesian Deep Tensor Factorization

Despite the elegance and convenience of the popular Tucker and CP factorization, their multilinear form can severely limit the capability of estimating complicated, highly nonlinear/nonstationary

relationships hidden in data. While numerous other methods have also been proposed, *e.g.,* (Chu & Ghahramani, 2009; Kang et al., 2012; Choi & Vishwanathan, 2014), most of them are still inherently based on the CP or Tucker form. Enlightened by the expressive power of (deep) neural networks (Goodfellow et al., 2016), we propose a Bayesian deep tensor factorization model to overcome the limitation of traditional methods and flexibly estimate all kinds of complex relationships.

Specifically, for each tensor entry $\mathbf{i}$, we construct an input $\mathbf{x_i}$ by concatenating all the embedding vectors associated with $\mathbf{i}$, namely, $\mathbf{x_i} = [(\mathbf{u}_{i_1}^1)^\top, \ldots, (\mathbf{u}_{i_K}^K)^\top]^\top$. We assume that there is an unknown mapping between the input embeddings $\mathbf{x_i}$ and the value of entry $\mathbf{i}$, $f : \mathbb{R}^{\sum_{k=1}^M r_k} \to \mathbb{R}$, which reflects the complex interactions/relationships between the tensor nodes in entry $\mathbf{i}$. Note that CP factorization uses a multilinear mapping. We use an $M$-layer neural network (NN) to model the mapping $f$, which are parameterized by $M$ weight matrices $\mathcal{W} = \{\mathbf{W}_1, \ldots, \mathbf{W}_M\}$. Each $\mathbf{W}_m$ is $V_m \times (V_{m-1} + 1)$ where $V_m$ and $V_{m-1}$ are the widths for layer $m$ and $m-1$, respectively; $V_0 = \sum_{k=1}^K r_k$ is the input dimension and $V_M = 1$. We denote the output in each hidden layer $m$ by $\mathbf{h}_m$ ($1 \le m \le M-1$) and define $\mathbf{h}_0 = \mathbf{x_i}$. We compute each $\mathbf{h}_m = \sigma(\mathbf{W}_m[\mathbf{h}_{m-1}; 1]/\sqrt{V_{m-1}+1})$ where $\sigma(\cdot)$ is a nonlinear activation function, *e.g.,* ReLU and tanh. Note that we append a constant feature 1 to introduce the bias terms in the linear transformation, namely the last column in each $\mathbf{W}_m$. For the last layer, we compute the output by $f_\mathcal{W}(\mathbf{x_i}) = \mathbf{W}_M[\mathbf{h}_{M-1}; 1]/\sqrt{V_{M-1}+1}$. Given the output, we sample the observed entry value $y_\mathbf{i}$ via a noisy model. For continuous data, we use a Gaussian noise model, $p(y_\mathbf{i}|\mathcal{U}) = \mathcal{N}(y_\mathbf{i}|f_\mathcal{W}(\mathbf{x_i}), \tau^{-1})$ where $\tau$ is the inverse noise variance. We further assign $\tau$ a Gamma prior, $p(\tau) = \text{Gamma}(\tau|a_0, b_0)$. For binary data, we use the Probit model, $p(y_\mathbf{i}|\mathcal{U}) = \Phi((2y_\mathbf{i} - 1)f_\mathcal{W}(\mathbf{x_i}))$ where $\Phi(\cdot)$ is the cumulative density function (CDF) of the standard normal distribution.

Despite their great flexibility, NNs take the risk of overfitting. The larger a network, *i.e.,* with more weight parameters, the easier the network overfits the data. In order to prevent overfitting, we assign a spike-and-slab prior (Ishwaran et al., 2005; Titsias & Lázaro-Gredilla, 2011) over each NN weight to sparsify and condense the network. Specifically, for each weight $w_{mjt} = [\mathbf{W}_m]_{jt}$, we first sample a binary selection indicator $s_{mjt}$ from $p(s_{mij}|\rho_0) = \text{Bern}(s_{mjt}|\rho_0) = \rho_0^{s_{mjt}}(1 - \rho_0)^{1-s_{mjt}}$. The weight is then sampled from

$$p(w_{mjt}|s_{mjt}) = s_{mjt}\mathcal{N}(w_{mjt}|0, \sigma_0^2) + (1 - s_{mjt})\delta(w_{mjt}), \tag{3}$$

where $\delta(\cdot)$ is the Dirac-delta function. Hence, the selection indicator $s_{mjt}$ determines the type of prior over $w_{mjt}$: if $s_{mjt}$ is 1, meaning the weight is useful and active, we assign a flat Gaussian prior with variance $\sigma_0^2$ (slab component); if otherwise $s_{mjt}$ is 0, namely the weight is useless and should be deactivated, we assign a spike prior concentrating on 0 (spike component).

Finally, we place a standard normal prior over the embeddings $\mathcal{U}$. Given the set of observed tensor entries $\mathcal{D} = \{y_{\mathbf{i}_1}, \ldots, y_{\mathbf{i}_N}\}$, the joint probability of our model for continuous data is

$$p(\mathcal{U}, \mathcal{W}, \mathcal{S}, \tau) = \prod_{m=1}^M \prod_{j=1}^{V_m} \prod_{t=1}^{V_{m-1}+1} \text{Bern}(s_{mjt}|\rho_0)(s_{mjt}\mathcal{N}(w_{mjt}|0, \sigma_0^2) + (1 - s_{mjt})\delta(w_{mjt}))$$

$$\cdot \prod_{k=1}^K \prod_{j=1}^{d_k} \mathcal{N}(\mathbf{u}_j^k|\mathbf{0}, \mathbf{I})\text{Gamma}(\tau|a_0, b_0) \prod_{n=1}^N \mathcal{N}(y_{\mathbf{i}_n}|f_\mathcal{W}(\mathbf{x}_{\mathbf{i}_n}), \tau^{-1}) \tag{4}$$

where $\mathcal{S} = \{s_{mjt}\}$, and for binary data is

$$p(\mathcal{U}, \mathcal{W}, \mathcal{S}) = \prod_{m=1}^M \prod_{j=1}^{V_m} \prod_{t=1}^{V_{m-1}+1} \text{Bern}(s_{mjt}|\rho_0)(s_{mjt}\mathcal{N}(w_{mjt}|0, \sigma_0^2) + (1 - s_{mjt})\delta(w_{mjt}))$$

$$\cdot \prod_{k=1}^K \prod_{j=1}^{d_k} \mathcal{N}(\mathbf{u}_j^k|\mathbf{0}, \mathbf{I}) \prod_{n=1}^N \Phi((2y_{\mathbf{i}_n} - 1)f_\mathcal{W}(\mathbf{x}_{\mathbf{i}_n})). \tag{5}$$

## 4   Streaming Posterior Inference

We now present our streaming model estimation algorithm. In general, the observed tensor entries are assumed to be streamed in a sequence of small batches, $\{\mathcal{B}_1, \mathcal{B}_2, \ldots\}$. Different batches do not have to include the same number of entries. Upon receiving each batch $\mathcal{B}_t$, we aim to update the posterior distribution of the embeddings $\mathcal{U}$, the inverse noise variance $\tau$ (for continuous data),

the selection indicators $\mathcal{S}$ and the neural network weights $\mathcal{W}$, without re-accessing the previous batches $\{\mathcal{B}_j\}_{j<t}$. While we can apply SVB, the variational ELBO that integrates the current posterior and the new entry batch will be analytically intractable. Take binary tensors as an example. Given a new entry batch $\mathcal{B}_t$, the EBLO constructed based on the blending distribution (see (2)) is $\mathcal{L} = -\mathrm{KL}\big(q(\mathcal{U},\mathcal{S},\mathcal{W})\|q_{\mathrm{cur}}(\mathcal{U},\mathcal{S},\mathcal{W})\big) + \sum_{n\in\mathcal{B}_t}\mathbb{E}_q\big[\log\Phi\big((2y_{\mathbf{i}_n}-1)f_{\mathcal{W}}(\mathbf{x}_{\mathbf{i}_n})\big)\big]$. Due to the nested, nonlinear coupling of the embeddings (in each $\mathbf{x}_{\mathbf{i}_n}$) and NN weights $\mathcal{W}$ in calculating $f_{\mathcal{W}}(\mathbf{x}_{\mathbf{i}_n})$, the expectation terms in $\mathcal{L}$ are intractable, without any closed form. Obviously, the same conclusion applies to the continuous data. Therefore, to maximize $\mathcal{L}$ so as to obtain the updated posterior, we have to use stochastic gradient descent (SGD), typically with the re-parameterization trick (Kingma & Welling, 2013). However, without the explicit form of $\mathcal{L}$, it is hard to diagnosis the convergence of SGD — it may stop at a place far from the (local) optimums. Note that we cannot use hold-out data or cross-validation to monitor/control the training because we cannot store or revisit data. The inferior posterior estimation in one batch can in turn influence the posterior updates in the subsequent batches, and finally result in a very poor model estimation.

## 4.1 Online Moment Matching for Posterior Update

To address these problems, we exploit the assumed-density-filtering (ADF) framework (Boyen & Koller, 1998), which can be viewed as an online version of expectation propagation (EP) (Minka, 2001a), a general approximate Bayesian inference algorithm. ADF is also based on the incremental version of Bayes' rule (see (1)). It uses a distribution in the exponential family (Wainwright et al., 2008) to approximate the current posterior. When the new data arrive, instead of maximizing a variational ELBO, ADF projects the (unnormalized) blending distribution (2) to the exponential family to obtain the updated posterior. The projection is done by moment matching, which essentially is to minimize $\mathrm{KL}(\tilde{p}(\boldsymbol{\theta})/Z\|q(\boldsymbol{\theta}))$ where $Z$ is the normalization constant. For illustration, suppose we choose $q(\boldsymbol{\theta})$ to be a fully factorized Gaussian distribution, $q(\boldsymbol{\theta}) = \prod_j q(\theta_j) = \prod_j \mathcal{N}(\theta_j|\mu_j, v_j)$. To update each $q(\theta_j)$, we compute the first and second moments of $\theta_j$ w.r.t $\tilde{p}(\boldsymbol{\theta})$, and match a Gaussian distribution with the same moments, namely, $\mu_j = \mathbb{E}_{\tilde{p}}(\theta_j)$ and $v_j = \mathrm{Var}_{\tilde{p}}(\theta_j) = \mathbb{E}_{\tilde{p}}(\theta_j^2) - \mathbb{E}_{\tilde{p}}(\theta_j)^2$.

For our model, we use a fully factorized distribution in the exponential family to approximate the current posterior. When a new batch of data $\mathcal{B}_t$ are received, we sequentially process each observed entry, and perform moment matching to update the posterior of the NN weights $\mathcal{W}$ and associated embeddings. Specifically, let us start with the binary data. We approximate the posterior with

$$q_{\mathrm{cur}}(\mathcal{W},\mathcal{U},\mathcal{S}) = \prod_{m=1}^{M}\prod_{j=1}^{V_m}\prod_{t=1}^{V_{m-1}+1}\mathrm{Bern}(s_{mjt}|\rho_{mjt})\mathcal{N}(w_{mjt}|\mu_{mjt}, v_{mjt})\prod_{k=1}^{K}\prod_{j=1}^{d_k}\prod_{t=1}^{r_k}\mathcal{N}(u_j^k|\psi_{kjt}, \nu_{kjt}).$$

Given each entry $\mathbf{i}_n$ in the new batch, we construct the blending distribution, $\tilde{p}(\mathcal{W},\mathcal{U},\mathcal{S}) \propto q_{\mathrm{cur}}(\mathcal{W},\mathcal{U},\mathcal{S})\Phi\big((2y_{\mathbf{i}_n}-1)f_{\mathcal{W}}(\mathbf{x}_{\mathbf{i}_n})\big)$. To obtain its moments, we consider the normalizer, *i.e.*, the model evidence under the blending distribution,

$$Z_n = \int q_{\mathrm{cur}}(\mathbf{W},\mathcal{U},\mathcal{S})\Phi\big((2y_{\mathbf{i}_n}-1)f_{\mathcal{W}}(\mathbf{x}_{\mathbf{i}_n})\big)\mathrm{d}\mathcal{W}\mathrm{d}\mathcal{U}\mathrm{d}\mathcal{S}. \tag{6}$$

Under the Gaussian form, according to (Minka, 2001b), we can compute the moments and update the posterior of each NN weight $w_{mjt}$ and each embedding element associated with $\mathbf{i}_n$— $\{u_{i_{nk}}^k\}_k$ by

$$\mu^* = \mu + v\frac{\partial\log Z_n}{\partial\mu}, \quad v^* = v - v^2\Big[\big(\frac{\partial\log Z_n}{\partial\mu}\big)^2 - 2\frac{\partial\log Z_n}{\partial v}\Big], \tag{7}$$

where $\mu$ and $v$ are the current posterior mean and variance of the corresponding weight or embedding element. Note that since the likelihood does not include the binary selection indicators $\mathcal{S}$, their moments are the same as those under $q_{\mathrm{cur}}$ and we do not need to update their posterior.

However, a critical issue is that due to the nonlinear coupling of the $\mathcal{U}$ and $\mathcal{W}$ in computing the NN output $f_{\mathcal{W}}(\mathbf{x}_{\mathbf{i}_n})$, the exact normalizer is analytically intractable. To overcome this issue, we consider approximating the current posterior of $f_{\mathcal{W}}(\mathbf{x}_{\mathbf{i}_n})$ first. We use multivariate Delta's method (Oehlert, 1992; Bickel & Doksum, 2015) that expands the NN output at the mean of $\mathcal{W}$ and $\mathcal{U}$,

$$f_{\mathcal{W}}(\mathbf{x}_{\mathbf{i}_n}) \approx f_{\mathbb{E}[\mathcal{W}]}(\mathbb{E}[\mathbf{x}_{\mathbf{i}_n}]) + \mathbf{g}_n^{\top}(\boldsymbol{\eta}_n - \mathbb{E}[\boldsymbol{\eta}_n]) \tag{8}$$

where the expectation is under $q_{\mathrm{cur}}(\cdot)$, $\boldsymbol{\eta}_n = \mathrm{vec}(\mathcal{W}\cup\mathbf{x}_{\mathbf{i}_n})$, $\mathbf{g}_n = \nabla f_{\mathcal{W}}(\mathbf{x}_{\mathbf{i}_n})|_{\boldsymbol{\eta}_n=\mathbb{E}[\boldsymbol{\eta}_n]}$. Note that $\mathbf{x}_{\mathbf{i}_n}$ is the concatenation of the embeddings associated with $\mathbf{i}_n$. The rationale of the approximation

(8) is that the NN output is highly nonlinear and nonconvex to $\mathcal{U}$ and $\mathcal{W}$. Hence, the scale of the output change rate (*i.e.,* gradient) can be much larger than the scale of the posterior variances of $\mathcal{W}$ and $\mathcal{U}$, which are (much) smaller than prior variance 1 (see (4) and (5)). Therefore, we can ignore the second-order term and just use the first-order Taylor approximation. We have also tried the second-order expansion, which, however, is unstable and does not improve the performance.

Based on (8), we can calculate the first and second moments of $f_\mathcal{W}(\mathbf{x}_{\mathbf{i}_n})$,

$$\alpha_n = \mathbb{E}_{q_{\text{cur}}}[f_\mathcal{W}(\mathbf{x}_{\mathbf{i}_n})] \approx f_{\mathbb{E}[\mathcal{W}]}(\mathbb{E}[\mathbf{x}_{\mathbf{i}_n}]), \ \ \beta_n = \text{Var}_{q_{\text{cur}}}(f_\mathcal{W}(\mathbf{x}_{\mathbf{i}_n})) \approx \mathbf{g}_n^\top \text{diag}(\boldsymbol{\gamma}_n)\mathbf{g}_n \quad (9)$$

where each $[\boldsymbol{\gamma}_n]_j = \text{Var}_{q_{\text{cur}}}([\boldsymbol{\eta}_n]_j)$. Due to the fully factorized posterior form, we have $\text{cov}(\boldsymbol{\eta}_n) = \text{diag}(\boldsymbol{\eta}_n)$. Note that all the information in the Gaussian posterior (*i.e.,* mean and variance) of $\mathcal{W}$ and $\mathcal{U}$ have been integrated to approximate the moments and posterior of the NN output. Now we use moment matching to approximate the current (marginal) posterior of the NN output by $q_{\text{cur}}(f_\mathcal{W}(\mathbf{x}_{\mathbf{i}_n})) = \mathcal{N}(f_\mathcal{W}(\mathbf{x}_{\mathbf{i}_n})|\alpha_n, \beta_n)$. Then we compute the running model evidence (6) by

$$Z_n = \mathbb{E}_{q_{\text{cur}}(\mathcal{W},\mathcal{U},\mathcal{S})}[\Phi\big((2y_{\mathbf{i}_n} - 1)f_\mathcal{W}(\mathbf{x}_{\mathbf{i}_n})\big)] = \mathbb{E}_{q_{\text{cur}}(f_\mathcal{W}(\mathbf{x}_{\mathbf{i}_n}))}[\Phi\big((2y_{\mathbf{i}_n} - 1)f_\mathcal{W}(\mathbf{x}_{\mathbf{i}_n})\big)]$$

$$\approx \int \mathcal{N}(f_o|\alpha_n, \beta_n)\Phi\big((2y_{\mathbf{i}_n} - 1)f_o\big)\text{d}f_o = \Phi\big(\frac{(2y_{\mathbf{i}_n} - 1)\alpha_n}{\sqrt{1 + \beta_n}}\big) \quad (10)$$

where we redefine $f_o = f_\mathcal{W}(\mathbf{x}_{\mathbf{i}_n})$ for simplicity. With the nice analytical form, we can immediately apply (7) to update the posterior for $\mathcal{W}$ and the associated embeddings in $\mathbf{i}_n$. In light of the NN structure, the gradient can be efficiently calculated via back-propagation, which can be automatically done by many deep learning libraries.

For continuous data, we introduce a Gamma posterior for the inverse noise variance, $q_{\text{cur}}(\tau) = \text{Gamma}(\tau|a, b)$, in addition to the fully factorized posterior for $\mathcal{W}, \mathcal{U}$ and $\mathcal{S}$ as in the binary case. After we use (8) and (9) to obtain the posterior of the NN output, we derive the running model evidence by

$$Z_n = \mathbb{E}_{q_{\text{cur}}(\mathcal{W},\mathcal{U},\mathcal{S},\tau)}[\mathcal{N}\big(y_{\mathbf{i}_n}|f_\mathcal{W}(\mathbf{x}_{\mathbf{i}_n}), \tau^{-1}\big)] = \mathbb{E}_{q_{\text{cur}}(f_o)q_{\text{cur}}(\tau)}[\mathcal{N}(y_{\mathbf{i}_n}|f_o, \tau^{-1})]$$

$$\approx \mathbb{E}_{q_{\text{cur}}(\tau)}\big[\int \mathcal{N}(f_o|\alpha_n, \beta_n)\mathcal{N}(y_{\mathbf{i}_n}|f_o, \tau^{-1})\text{d}f_o\big] = \mathbb{E}_{q_{\text{cur}}(\tau)}[\mathcal{N}(y_{\mathbf{i}_n}|\alpha_n, \beta_n + \tau^{-1})]. \quad (11)$$

Next, we use the first-order Taylor expansion again, at the mean of $\tau$, to approximate the Gaussian term inside the expectation, $\mathcal{N}(y_{\mathbf{i}_n}|\alpha_n, \beta_n + \tau^{-1}) \approx \mathcal{N}(y_{\mathbf{i}_n}|\alpha_n, \beta_n + \mathbb{E}_{q_{\text{cur}}}[\tau]^{-1}) + (\tau - \mathbb{E}_{q_{\text{cur}}}[\tau])\partial\mathcal{N}(y_{\mathbf{i}_n}|\alpha_n, \beta_n + \tau^{-1})/\partial\tau|_{\tau=\mathbb{E}_{q_{\text{cur}}}[\tau]}$. Taking expectation over the Taylor expansion gives

$$Z_n \approx \mathcal{N}(y_{\mathbf{i}_n}|\alpha_n, \beta_n + \mathbb{E}_{q_{\text{cur}}}(\tau)^{-1}) = \mathcal{N}(y_{\mathbf{i}_n}|\alpha_n, \beta_n + b/a). \quad (12)$$

We now can use (7) to update the posterior of $\mathcal{W}$ and the embeddings associated with the entry. While we can also use more accurate approximations, *e.g.,* the second-order Taylor expansion or quadrature, we found empirically our method achieves almost the same performance.

To update $q_{\text{cur}}(\tau)$, we consider the blending distribution only in terms of the NN output $f_o$ and $\tau$ so we have $\tilde{p}(f_o, \tau) \propto q_{\text{cur}}(f_o)q_{\text{cur}}(\tau)\mathcal{N}(y_{\mathbf{i}_n}|f_o, \tau^{-1}) = \mathcal{N}(f_o|\alpha_n, \beta_n)\text{Gamma}(\tau|a, b)\mathcal{N}(y_{\mathbf{i}_n}|f_o, \tau^{-1})$. Then we follow (Wang & Zhe, 2019) to first derive the conditional moments and then approximate the expectation of the conditional moments to obtain the moments. The details are given in the supplementary material. The updated posterior is given by $q^*(\tau) = \text{Gamma}(\tau|a^*, b^*)$, where $a^* = a + \frac{1}{2}$ and $b^* = b + \frac{1}{2}((y_{\mathbf{i}_n} - \alpha_n)^2 + \beta_n)$.

### 4.2 Prior Approximation Refinement

Ideally, at the beginning (*i.e.,* when we have not received any data), we should set $q_{\text{cur}}$ to the prior of the model (see (4) and (5)). This is feasible for the embeddings $\mathcal{U}$, selection indicators $\mathcal{S}$ and the inverse noise variance $\tau$ (for continuous data only), because their Gaussian, Bernoulli and Gamma priors are all members of the exponential family. However, the spike-and-slab prior for each NN weight $w_{mjt}$ (see (3)) is a mixture prior and does not belong to the exponential family. Hence, we introduce an approximation term,

$$p(w_{mjt}|s_{mjt}) \overset{\propto}{\sim} A(w_{mjt}, s_{mjt}) = \text{Bern}\big(s_{mjt}|c(\rho_{mjt})\big)\mathcal{N}(w_{mjt}|\mu^0_{mjt}, v^0_{mjt}) \quad (13)$$

where $\overset{\propto}{\sim}$ means "approximately proportional to" and $c(x) = 1/(1 + \exp(-x))$. At the beginning, we initialize $v^0_{mjt} = \sigma^2_0$ and $\mu^0_{mjt}$ to be a random number generated from a standard Gaussian distribution

truncated in $[-\sigma_0, \sigma_0]$; we initialize $\rho_{mjt} = 0$. Obviously, this is a very rough approximation. If we only execute the standard ADF to continuously integrate new entries to update the posterior (see (7)), the prior approximation term will remain the same and never be changed. However, the spike-and-slab prior is critical to sparsify and condense the network, and an inferior approximation will make it noneffective at all. To address this issue, after we process all the entries in the incoming batch, we use EP to update/improve the prior approximation term (13), with which to further update the posterior of the NN weights. In this way, as we process more and more batches of the observed tensor entries, the prior approximation becomes more and more accurate, and thereby can effectively inhibit/deactivate the redundant or useless weights on the fly. The details of the updates are provided in the supplementary material. Finally, we summarize our streaming inference in Algorithm 1 in the supplementary material.

### 4.3 Algorithm Complexity

The time complexity of our streaming inference is $\mathcal{O}(NV + \sum_{k=1}^{K} d_k r_k)$ where $V$ is the total number of weights in the NN. Therefore, the computational cost is proportional to $N$, the size of the streaming batch. The space complexity is $\mathcal{O}(V + \sum_{k=1}^{K} d_k r_k)$, including the storage of the posterior for the embeddings $\mathcal{U}$, NN weights $\mathcal{W}$ and selection indicators $\mathcal{S}$, and the approximation term for the spike-and-slab prior.

## 5 Related Work

Classical CP (Harshman, 1970) and Tucker (Tucker, 1966) factorization are multilinear and therefore are incapable of estimating complex, nonlinear relationships in data. While numerous other approaches have been proposed (Shashua & Hazan, 2005; Chu & Ghahramani, 2009; Sutskever et al., 2009; Hoff, 2011; Kang et al., 2012; Yang & Dunson, 2013; Choi & Vishwanathan, 2014; Hu et al., 2015; Rai et al., 2015), most of them are still based on the CP or Tucker form. To overcome these issues, recently, several Bayesian nonparametric tensor factorization models (Xu et al., 2012; Zhe et al., 2015; 2016) were proposed to estimate the nonlinear relationships with Gaussian processes (Rasmussen & Williams, 2006). The most recent work, NeurlCP (Liu et al., 2018) and CoSTCo (Liu et al., 2019) have shown the advantage of NNs in tensor factorization. CoSTCo also concatenates the embeddings associated with each entry to construct the input and use the NN output to predict the entry value; but to alleviate overfitting, CoSTCo introduces two convolutional layers to extract local features and then feed them into dense layers. By contrast, with the spike-and-slab prior and Bayesian inference, we found that our model can also effectively prevent overfitting, without the need for extra convolutional layers. NeuralCP uses two NNs, one to predict the entry value, the other the (log) noise variance. Hence, NeuralCP only applies to continuous data. Our model can be used for both continuous and binary data. Finally, both CoSTCo and NerualCP are trained with stochastic optimization, need to pass the data many times (epochs), and hence cannot handle streaming data.

Expectation propagation (Minka, 2001a) is an approximate Bayesian inference algorithm that generalizes assumed-density-filtering (ADF) (Boyen & Koller, 1998) and (loopy) belief propagation (Murphy et al., 1999). EP employs an exponential-family term to approximate the prior and likelihood of each data point, and cyclically updates each approximation term via moment matching. ADF can be considered as applying EP in a model including only one data point. Because ADF only maintains the holistic posterior, without the need for keeping individual approximation terms, it is very appropriate for streaming learning. EP can meet a practical barrier when the moment matching is intractable. To address this problem, Wang & Zhe (2019) proposed conditional EP that uses conditional moment matching, quadrature and Taylor approximations to provide a high-quality, analytical solution. Based on EP and ADF, Hernández-Lobato & Adams (2015) proposed probabilistic back-propagation, a batch inference algorithm for Bayesian neural networks. PBP conducts ADF to pass the dataset many times and re-update the prior approximation after each pass. A key difference from our work is that PBP conducts a forward, layer by layer moment matching, to approximate the posterior of each hidden neuron in the network, until it reaches the output. The computation is limited to fully connected, feed-forward networks and $\mathrm{ReLU}$ activation function. By contrast, our method computes the moments of the NN output via Delta's method (*i.e.,* Taylor expansions) and does not need to approximate the posterior of the hidden neurons. Therefore, our method is free to use any NN architecture and activation function. Note that multi-variate Delta's method was also used in Laplace's approximation (MacKay, 1992) and non-conjugate variational inference Wang & Blei (2013). Furthermore, we employ spike-and-slab priors over the NN weights to control the complexity of the model and to prevent overfitting in the streaming inference.

# 6 Experiment

## 6.1 Predictive Performance

**Datasets**. We examined SPIDER on four real-world, large-scale datasets. (1) *DBLP* (Du et al., 2018), a binary tensor about bibliography relationships *(author conference, keyword)*, of size $10,000 \times 200 \times 10,000$, including $0.001\%$ nonzero entries. (2) *Anime*(https://www.kaggle.com/CooperUnion/anime-recommendations-database), a two-mode tensor depicting binary *(user, anime)* preferences. The tensor contains $1,300,160$ observed entries, of size $25,838 \times 4,066$. (3) *ACC* (Du et al., 2018), a continuous tensor representing the three-way interactions *(user, action, file)*, of size $3,000 \times 150 \times 30,000$, including $0.9\%$ nonzero entries. (4) *MovieLen1M* (https://grouplens.org/datasets/movielens/), a two-mode continuous tensor of size $6,040 \times 3,706$, consisting of *(user, movie)* ratings. We have $1,000,209$ observed entries.

**Competing methods.** We compared with the following baselines. (1) POST (Du et al., 2018), the state-of-the-art probabilistic streaming tensor decomposition algorithm based on the CP model. It uses streaming variational Bayes (SVB) (Broderick et al., 2013) to perform mean-field posterior updates upon receiving new entries. (2) SVB-DTF, SVB based deep tensor factorization. (3) SVB-GPTF, the streaming version of the Gaussian process(GP) based nonlinear tensor factorization (Zhe et al., 2016), implemented with SVB. Note that similar to NNs, the ELBO in SVB for GP factorization is intractable and we used stochastic optimization. (4) SS-GPTF, the streaming GP factorization implemented with the recent streaming sparse GP approximations (Bui et al., 2017). It uses SGD to optimize another intractable ELBO. (5) CP-WOPT (Acar et al., 2011a), a scalable static CP factorization algorithm implemented with gradient-based optimization.

**Parameter Settings.** We implemented our method, SPIDER with Theano, and SVB-DTF, SVB/SS-GPTF with TensorFlow. For POST, we used the original MATLAB implementation (https://github.com/yishuaidu/POST). For SVB/SS-GPTF, we set the number of pseudo inputs to 128 in their sparse approximations. We used Adam (Kingma & Ba, 2014) for the stochastic optimization in SVB-DTF and SVB/SS-GPTF, where we set the number of epochs to 100 in processing each streaming batch and tuned the learning rate from $\{10^{-5}, 5 \times 10^{-5}, 10^{-4}, 3 \times 10^{-4}, 5 \times 10^{-4}, 10^{-3}, 3 \times 10^{-3}, 5 \times 10^{-3}, 10^{-2}\}$. For SPIDER and SVB-DTF, We used a 3-layer NN, with 50 nodes in each hidden layer. We tested ReLU and tanh activations.

We first evaluated the prediction accuracy after all the (accessible) entries are processed. To do so, we sequentially fed the training entries into every method, each time a small batch. We then evaluated the predictive performance on the test entries. We examined the root-mean-squared-error (RMSE) and area under ROC curves (AUC) for continuous and binary data, respectively. We ran the static factorization algorithm CP-WOPT on the entire training set. On *DBLP* and *ACC*, we used the same split of the training and test entries in (Du et al., 2018), including 320K and 1M training entries for *DBLP* and *ACC* respectively, and 100K test entries for both. On *Anime* and *MovieLen1M*, we randomly split the observed entries into 90% for training and 10% for test. For both datasets, the number of training entries is around one million. For each streaming factorization approach, we randomly shuffled the training entries and then partitioned them into a stream of batches. On each dataset, we repeated the test for 5 times and calculated the average of RMSEs/AUCs and their standard deviations. For CP-WOPT, we used different random initializations in each test.

We conducted two groups of evaluations. In the first group, we fixed the batch size to 256 and examined the predictive performance with different ranks (or dimensions) of the embedding vectors, $\{3, 5, 8\ 10\}$. In the second group, we fixed the rank to 8, and examined how the prediction accuracy varies with the size of the streaming batches, $\{2^6, 2^7, 2^8, 2^9\}$. The results are reported in Fig. 1. As we can see, in both settings, our method SPIDER (with both tanh and ReLU) consistently outperforms all the competing approaches in all the cases and mostly by a large margin. First, SPIDER significantly improves upon POST and CP-WOPT — the streaming and static multilinear factorization, confirming the advantages of the deep tensor factorization. It worth noting that CP-WOPT performed much worse than POST on *ACC* and *MovieLen1M*, which might be due to the poor local optimums CP-WOPT converged to. Second, SVB/SS-GPTF are generally far worse than our method, and in many cases even inferior to POST (see Fig. 1b, c, d and f). SVB-DTF are even worse. Only on Fig. 1c, the performance of SVB-DTF is comparable to or better than CP-WOPT, and in all the other cases, SVB-DTF is much worse than all the other methods and we did not show its curve in the figure (similar for CP-WOPT in Fig. 1g). Those results might be due to the inferior/unreliable stochastic posterior updates. Lastly, although both SPIDER-ReLU and SPIDER-tanh outperform all the baselines, SPIDER-ReLU is overall better than SPIDER-tanh (especially Fig. 1g).

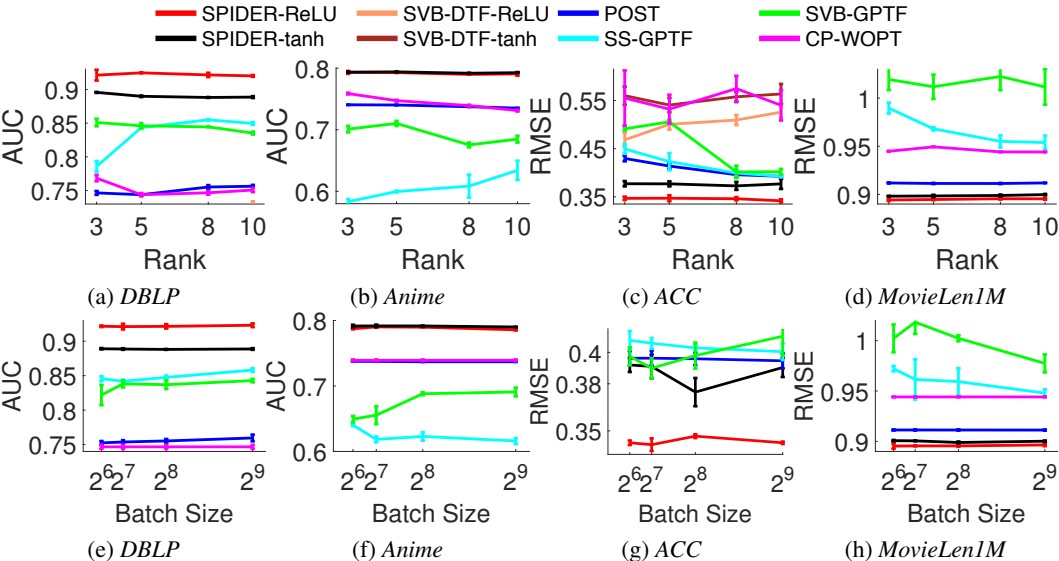

Figure 1: Predictive performance with different ranks (top row) and streaming batch sizes (bottom row). In the top row, the streaming bath size is fixed to 256; in the bottom row, the rank is fixed to 8. The results are

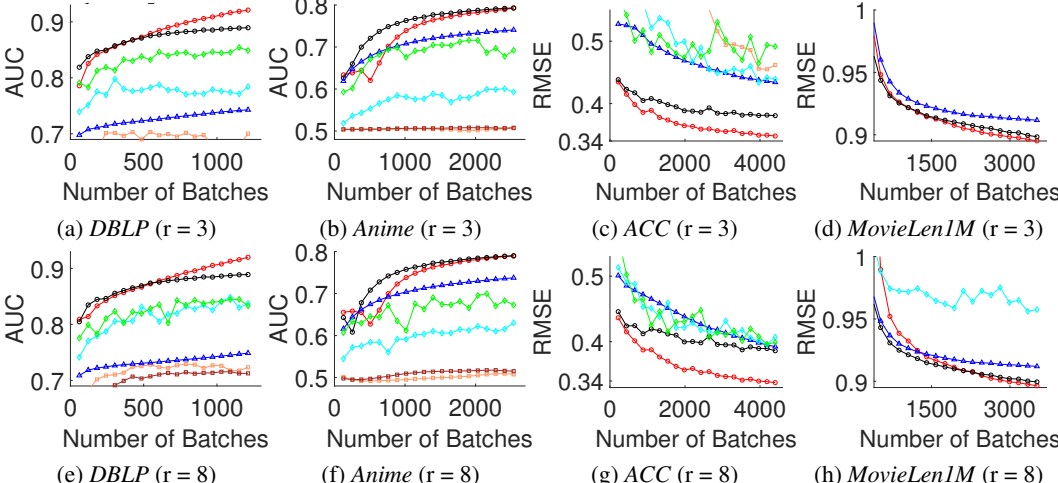

Figure 2: Running prediction accuracy along with the number of processed streaming batches. The batch size was fixed to 256.

## 6.2 Prediction On the Fly

Next, we evaluated the dynamic performance. We randomly generated a stream of training batches from each dataset, upon which we ran each algorithm and examined the prediction accuracy after processing each batch. We set the batch size to 256 and tested with rank $r = 3$ and $r = 8$. The running RMSE/AUC of each method are reported in Fig. 2. Note that some curves are missing or partly missing because the performance of the corresponding methods are much worse than all the other ones. In general, nearly all the methods improved the prediction accuracy with more and more batches, showing increasingly better embedding estimations. However, SPIDER always obtained the best AUC/RMSE on the fly, except at the beginning stage on *Anime* and *MovieLen1M* ($r = 8$). In addition, the trends of SPIDER and POST are much smoother than that of SVB-DTF and SVB/SS-GPTF, which might again because the stochastic updates in the latter methods are unstable and unreliable. Note that in Fig. 2b, SVB-DTF has running AUC steadily around 0.5, implying that SVB actually failed to effectively update the posterior. Finally, in the supplementary material, we provide the running time, and showcase the sparse posteriors of the NN weights learned by SPIDER.

## 7 Conclusion

We have presented SPIDER, a streaming probabilistic deep tensor factorization approach, which can effectively leverage neural networks to capture complicated relationships for streaming factorization. Experiments on real-world applications have demonstrated the advantage of SPIDER.

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
