# OpenReview forum: "Streaming Probabilistic Deep Tensor Factorization"
_ICLR.cc/2021/Conference — Reject_

### Official Review · AnonReviewer2 · 2020-10-27
**Reasonable model and good experimental results**

**Rating:** 6
**Confidence:** 3

**Review:**

##########################

After author feedback:
Thanks for the detailed feedback from the authors. Most of my concerns have been addressed and I will keep my scores unchanged. Please add the additional information in the feedback to the final version.

##########################

Summary:

This paper proposes a streaming probabilistic deep tensor factorization model, called SPIDER, to solve the tensor factorization problem under streaming setting with deep factorization parameterized by Bayesian neural networks. To encourage sparsity and prevent overfitting, a spike-and-slab prior is used for the weights of the neural network layers. Learning algorithms are also derived based on the expectation propagation framework where the posterior of the network output is approximated using multivariate Delta's method and moment matching. Experimental results show improvement against existing tensor factorization models.


##########################

Pros:

1. Although there has been some work on streaming or online tensor factorization, the (Bayesian) deep factorization has never been addressed in the streaming setting, which is also quite an important problem to study.
2. The incorporation of spike-and-slab prior for sparsity is good for many practical applications.
3. Overall, the paper is organized and easy to follow.
4. The experiment section is well structured and multiple real-world large datasets are used.


##########################

Cons/Questions:

1. Does the algorithm converge? It would be great to see any theoretical or empirical analysis on convergence.
2. Two important baselines NeuralCP and CoSTCo are missing. As they are very recent and state-of-the-art models in neural tensor factorization, it would be great to see comparisons against them.
3. It is unclear from the experimental section that how many parameters does each model have. Can the authors provide a table or figure about the number of parameters of each model?
4. One of the most important properties of CP and Tucker decomposition is their interpretability. I am wondering if any interpretation of the factors can be carried out for the proposed model.
5. The caption of Fig. 1 seems unfinished or blocked by Fig 2.

---

> ### Author Response · Authors · 2020-11-14
> **Thanks for your comments.**
>
> C: comments; R: response
>
> C1: “Does the algorithm converge? It would be great to see any theoretical or empirical analysis on convergence”
>
> R1: Great question. Regarding the updates within each new batch of entries, our algorithm converges. Note that the updates of the Gaussian posterior in Eq. 7 and the s&s prior approximation Eq. 13 (see Sec. 2 in the supplementary material for details) are the analytical solution for minimizing the local KL divergence in the ADF/EP framework. After they are computed, the algorithm has already converged; we do not need to perform some iterative optimization. Regarding the updates after many streaming batches, the convergence is highly determined by the nature of the data. Many real-world applications keep generating new data, e.g., posts in Facebook and Youtube videos. The modals and patterns can be constantly changing over time. In such cases, we do not expect the posterior updates should converge, say, after receiving 10K entries. Instead, we expect our algorithm can continuously integrate the information of the new data and reflect the evolution of the data patterns. This is partially verified in Fig. 2 running accuracy. We can see that in DBLP, Anime and MovieLen1M dataset, the prediction accuracy keeps growing along with more streaming batches, and do not show convergence; only on ACC, it shows a trend of convergence (for tanh activation).
> Thanks for your great suggestion about performing further theoretical and empirical analysis of convergence. We will definitely conduct it in our future work.
>
> C2: “Two important baselines NeuralCP and CoSTCo are missing. As they are very recent and state-of-the-art models in neural tensor factorization, it would be great to see comparisons against them”
>
> R2: Great question. NerualCP and CoSTCo are not streaming factorization methods. They conduct traditional batch factorization, need to randomly sample minibatches of the observed entries, and run many epochs (i.e., passes) over the entire data. Therefore, these algorithms are NOT applicable to the streaming data, where the tensor entries are streamed in an order, only can be accessed once and are NOT allowed to store and revisit. For comparison, in our experiments, we have adjusted their SGD training plus reparameterization tricks to perform streaming factorization with deep neural networks (i.e., SVB-DTF-ReLU and SVB-DTF-tanh) and Gaussian processes (i.e., SVB-GPTF and SS-GPTF). See Fig. 1&2. However, we are happy to include their batch factorization performance to further enrich our experimental results.
>
> C3: “It is unclear from the experimental section that how many parameters does each model have. Can the authors provide a table or figure about the number of parameters of each model?”
>
> R3: Great idea! We will supplement this information in a table in our paper.
>
> C4: “One of the most important properties of CP and Tucker decomposition is their interpretability. I am wondering if any interpretation of the factors can be carried out for the proposed model.”
>
> R4: Great question! First, we can analyze the patterns exhibited in the learned embeddings. For example, we can run k-means or other clustering algorithms over the posterior means of the embeddings to find community structures and outliers. By collaborating with domain experts, e.g., in online advertising or recommendation, we can identify potential groups of customers with similar interests and anomaly users, which are important to design more accurate recommendation strategies and improve click-through rates. Second, we can contrast the prediction made by CP/Tucker decomposition and identify the cases where (1) the nonlinear prediction part (made by our model) is indeed the key to improve the accuracy, (2) the nonlinear prediction has the similar accuracy as the multilinear prediction, and (3) the nonlinear prediction hurts the prediction. We can analyze the distribution, the patterns and rationale for the three cases, which can provide a profound insight on the application. Third, we can look into the posterior variances of the embeddings and predictions, analyze when and why the estimation is very confident or uncertain, and how does it relate to data sparsity. These will be useful for decision making, e.g., ranking advertisements or products. Finally, we can use popular explanation tools, e.g., LIME [1], to extract more interpretations about the predictions made by nonlinear models.
>
> [1] Ribeiro, Marco Tulio, Sameer Singh, and Carlos Guestrin. "" Why should I trust you?" Explaining the predictions of any classifier." Proceedings of the 22nd ACM SIGKDD international conference on knowledge discovery and data mining. 2016.
>
> C5: “The caption of Fig. 1 seems unfinished or blocked by Fig 2”
>
> R5: Thanks for pointing out this issue. We will polish our paper to fix it.

---

### Official Review · AnonReviewer3 · 2020-10-28

**Rating:** 5
**Confidence:** 3

**Review:**



Update: The authors addressed part of my concerns. For the factor estimation, the proposed method relies on first order approximations while learning the posterior of the factors; however, the approximation error does not enter into the posterior.  The approximation also raises concerns regarding the convergence of the algorithm. Overall, I think the approach is promising, but some justification of the quality of the approximation is needed. Thus, I tend to keep my rating.

##############

The authors propose a streaming approach to tensor factorization with Bayesian neural networks. The problem is to factorize a three-way tensor with Gaussian noise. The proposed approach combines a Bayesian neural network (BNN) whose output predicts the entries of the tensor and the streaming variational Bayes (SVB) for incremental posterior updates. In addition, a spike-and-slab prior is placed on the BNN weights to encourage sparsity as well as prevent overfitting.  The authors performed empirical studies on four real datasets, DBLP, Anime, ACC, and MovieLen1M, where improved prediction accuracy of the tensor entries are reported.

From the empirical evaluation, the objective of the proposed approach is to ensure that the output of the BNN matches the tensor entries. However, this does not necessarily guarantee the correctness of the recovered factor matrices (input to the BNN). I feel that the problem setting of the paper is somehow different from the standard CP factorization setting where the (unique) factors are of primary interest. It would be good to add some discussions about the correctness and/or uniqueness of the uncovered factors.

Strengths:
- The idea of introducing a spike-and-slab prior in the factorization is interesting.
- Using SVB for online posterior updates is computationally practical.

Weaknesses:
- The presentation could be improved — it is not very clear how the factor matrices are estimated.
- Factorization using the proposed method is not unique (up to rescaling and permutation of the factor matrices).

---

> ### Author Response · Authors · 2020-11-14
> **Clarification of motivation and methods**
>
> C: comments; R: response
>
> C1: The uniqueness of factorization up to rescaling and permutation of the factor matrices.
>
> R1: We do agree that the classical CP factorization has the nice property that the factorization is unique. Even the matrix factorization A = PB does not have such property. We are willing to consider if we can put extra constraints to ensure some kinds of uniqueness.  However, our work focuses on discovering complex and nonlinear interactions or relationships from data, which are prevalent and critical in many real-world applications. The traditional multilinear methods, like CP and Tucker decomposition, however, are incompetent. This is also the motivation of the recent works of nonlinear factorization, for both matrices (e.g., GPLVM [1] and deep method (Xue, et al. 2017) mentioned by Reviewer 1) and tensors (Costco, NeurlCP, GPTF referenced in our paper; see Sec. 5 Related Work). In these works, learning a nonlinear representation to fully capture the data information is more important than achieving some elegant property, like uniqueness, by sacrificing the representation power.
>
> [1] Lawrence, Neil D. "Gaussian process latent variable models for visualisation of high dimensional data." Advances in neural information processing systems. 2004.
>
>
> C2: it is not very clear how the factor matrices are estimated.
>
> R2: We perform posterior inference. Therefore, we do NOT estimate the values of the factor matrices; instead, all the factors are viewed as random variables, and we estimate their posterior distributions in the streaming setting. The posterior of each factor is assumed to be Gaussian. When new tensor entries come in, the parameters of these Gaussians, i.e., the mean and variance, are updated by moment-matching in the ADF framework, which is equivalent to minimizing a local KL divergence (see Sec. 4.1, 1st paragraph for the details of ADF). The moment-matching is fulfilled by first computing the logarithm of the model evidence in Eq. 6 and then conducting a closed-form update in Eq. 7. Then we can obtain the new mean and variance of each Gaussian posterior. One of our key contributions is to effectively and efficiently compute the log evidence via Delta’s method. See the details in Sec. 4.1.

---

### Official Review · AnonReviewer4 · 2020-10-28
**Review of "STREAMING PROBABILISTIC DEEP TENSOR FACTORIZATION"**

**Rating:** 6
**Confidence:** 2

**Review:**

##################################################

After author feedback:

The authors have addressed my comments, though it is impossible to evaluate what the authors promise to do in future work. My evaluation remains unchanged.


##################################################

Summary:
The paper proposes a Bayesian neural network model for tensor factorization, with particular focus on streaming data. The method features spike-and-slab prior to avoid overfitting. The main computational challenge is how to handle the streaming data in the posterior inference steps. The authors propose an approximate algorithm via moment matching. The method shows promising results in simulation.

##################################################

Pros:
1. The proposed method is novel and mostly described and motivated in a clear fashion. It is an interesting addition to the current literature of Bayesian methods for streaming data.
2. The predictive performance, as shown by the authors, show significant improvements over the alternatives.

##################################################

Cons:
While I like the overall method, I have some questions about the inferential procedure. While it seems to work in the examples chosen by the authors, readers could benefit more from a clearer presentation of the methods, especially in terms of the computation and approximation accuracy trade-off.

1.  There are many (usually first order) approximation steps in the inference procedure. Can you quantify how these approximations affect the predictive performance and comment more on the validity of the Bayesian algorithm? To be more specific, I would like to see more justification or discussion in using the first order approximation of f_w(x_{i_n}) in both the binary and continuous case when calculating the normalizing constant Z_n. It is unclear how the approximation affects the estimation of Z_n after integration, and then in turn affects the posterior distribution. It also makes me wonder if the second-order Tayler expansion leads to no benefit in performance (claimed after equation 12) could be due to how f_w is approximated? The current justifications offered above equation 9 is useful in intuitively understand the considerations of the authors, yet not fully satisfactory why that is the right thing to do. In general, it is difficult for readers to see how the several approximations involved in the inference step change the posterior distribution computed versus the original target.

2. One thing that is missing is how the spike-and-slab selection indicators S are updated. Point mass spike are notoriously difficult to sample.  Is that the case here as well?

##################################################

Minor points:
1. Caption of figure 1 is cut out.
2. I can see the authors want to fit the length to 8 pages. Consider shotern the background section. Some of the notations can be introduced only when describing the proposed method.

---

> ### Author Response · Authors · 2020-11-14
> **Thanks for your comments**
>
> C: comments; R: response
>
> C1: Quantify the effects of the first-order approximation in the inference.
>
> R1: Great suggestions. Our approximation is referred to as (multi-variate) Delta’s method in statistical literature (see the references above eq. 8). The idea is to use the first-order Taylor approximation to calculate the moments of a nonlinear transformation of random variables. There have been a set of theories, based on Taylor’s theorem and asymptotic statistics (e.g., central limit theory), to characterize the error bound and convergence of this approximation. We will extend the existing theoretical framework to analyze the approximation errors in our model, how the errors propagate to the subsequent steps, e.g., log evidence calculation, error bound, etc.
>
> C2: One thing that is missing is how the spike-and-slab selection indicators S are updated. Point mass spike are notoriously difficult to sample. Is that the case here as well?
>
> R2: Thanks for the questions. Actually, we provide the details in Sec. 2 of the supplementary material, where Eq. 6 is the update regarding S. We mentioned this in Sec. 4.2 of the main paper. Although the spike component seems tricky for sampling, our updates are based on moment matching (to minimize a local KL divergence in EP/ADF framework, see Sec 4.1 1st paragraph for details) rather than Markov-Chain-Monte-Carlo (MCMC) sampling. Specifically, we view each variable in S as a random variable and update their posterior in the streaming setting. For each s_{mjt} in S, we assume a Bernoulli posterior, which is parameterized by \rho_{mjt} (see eq. 13). Note that while each s_{mjt} is binary, its posterior parameter \rho_{mjt} is still continuous. As shown in Eq. 6 of the supplementary material, the moment matching is analytical and tractable.

---

### Official Review · AnonReviewer1 · 2020-10-31
**Review for Streaming Probabilistic Deep Tensor Factorization**

**Rating:** 5
**Confidence:** 4

**Review:**

This paper proposes a probabilistic tensor factorization model for streaming data. The model uses:

* neural networks to learn richer factors,
* spike-and-slab prior on NN weights
* and online moment-matching for posterior inference.

Major comments:

1. Overall, the novelty of this work is minimal. There are existing works in literature on deep factorization models e.g. for recommender systems [Xue, et al. 2017]. It should be clarified how this work stands out.

2. Authors posit that existing tensor factorizations only conduct multilinear decompositions, yet to make inference of the proposed SPIDER model tractable, they use extensive linearization relaxations. How is the model capacity to learn non-linear relationships in data preserved despite these simplifications?

3. It has been claimed that stochastic gradient methods with re-parametrization trick perform poorly for parameter inference in this setting. It would be enlightening to see this is experimentally validated, as the SGD is the most successful optimization scheme for NN methods.

4. How does the spike-and-slab prior perform compared to other regularization methods such as drop-out?

5. The message of experiments section is not quite coherent. What is the trade-off between performance accuracy and computational complexity of the proposed framework?

---

> ### Author Response · Authors · 2020-11-14
> **Misunderstanding of the content and contributions of our work**
>
> We thank the reviewer for the comments and questions.  However, it seems that our paper is largely misunderstood, and we would like to clarify the content, contributions, and experimental details as follows. C: comments. R: response.
>
> C1: “Overall, the novelty of this work is minimal. There are existing works in literature on deep factorization models e.g. for recommender systems [Xue, et al. 2017]. It should be clarified how this work stands out.
>
> R1: First, we NEVER claim our model is the only deep tensor factorization (TF) model. We acknowledge other deep TF models (Costco and NeuralCP) at the beginning of our paper, see Sec. 1, 1st paragraph, line 6. In Sec. 5 (Related Work) 1st paragraph, we give a more detailed discussion.  While we are happy to also discuss and reference [Xue, et al. 2017], this work is about matrix factorization --- a different topic from our paper (tensor factorization).
>
> Second and more important, our key contribution is the streaming posterior inference of the deep TF models. To our knowledge, this is the FIRST work that enables highly flexible yet complex deep TF on streaming data. This is much more challenging than the traditional batch factorization, where we are free to sample mini-batches and run SGD for many epochs. In the streaming setting, all the data points are streamed in an order, can only be seen once, and never be revisited. See our motivation in 2nd paragraph of Sec. 1 (introduction). See the problem setting in Sec. 4, 1st paragraph.
>
> Third, the techniques we developed to fulfill deep streaming Bayesian TF, including using Taylor approximation + moment matching to approximate the log evidence, and approximating the bi-modal s&s prior to enable sparse streaming learning, have never been proposed in other streaming Bayesian models and applications. Therefore, we believe our motivation is strong and our work is novel enough.
>
> C2: “Authors posit that existing tensor factorizations only conduct multilinear decompositions, yet to make inference of the proposed SPIDER model tractable, they use extensive linearization relaxations. How is the model capacity to learn non-linear relationships in data preserved despite these simplifications?
>
> R2:  First, we NEVER claim that “existing tensor factorizations only conduct multilinear decompositions”. We do acknowledge the existing deep factorization work. See Sec. 1, 1st paragraph, line 6; see Sec. 5 (Related Work). In our experiments, we compared with state-of-the-art nonlinear TF models (see Sec. 6.1).
>
> Second, linearization relaxation in our inference does NOT imply any linear modeling assumption or linear parameter updates. The first-order Taylor expansion (eq. 8) is still highly nonlinear to the NN weights and embeddings. The posterior update (eq. 7) after the moment matching (eq. 9) is also nonlinear. In our paper (see Page 5, 1st paragraph), we have explained the rational why we can use this approximation. We have also found empirically the first-order approximation gives better and more stable performance than the second-order Taylor approximations.
>
> Third, our approximation is justifiable. In statistics, this is called (multivariate) Delta’s method, with rigorous analysis and asymptotic error bounds. see the reference in our paper (above eq. 8).
>
> C3: “It has been claimed that stochastic gradient methods with re-parametrization trick perform poorly for parameter inference in this setting. It would be enlightening to see this is experimentally validated, as the SGD is the most successful optimization scheme for NN methods.”
>
> R3: We did compare our method with SGD + re-parameterization trick in the streaming setting. The corresponding baselines are SVB-DTF-ReLU, SVB-DTF-tanh, SVB-GPTF, and SS-GPTF. SVB-DTF-ReLU and SVB-DTF-tanh use SGD+re-parameterization trick to perform streaming deep tensor factorization, while SS-GPTF and SVB-GPTF use these to perform nonlinear factorization based on Gaussian processes. See Sec. 2, Sec. 4 (1st paragraph) and Sec. 6.1 for details.
>
> C4: “How does the spike-and-slab prior perform compared to other regularization methods such as drop-out?
>
> R4: We did test drop-out along with SGD in our streaming applications. However, the performance is even worse than SGD and we did not report it in our figures. We will mention this in our paper.
>
> C5: “What is the trade-off between performance accuracy and computational complexity of the proposed framework?
>
> R5: The rank indicates computational complexity. The larger the rank, the more costly the computation. In Fig. 1a-d, we have shown how the prediction accuracy (in terms of AUC and RMSE) along with the rank in the streaming applications.  In Fig.2, we have also shown how the running prediction accuracy varies under different ranks (r=3 and r=8). Finally, we report the running time in Sec. 4 in the supplementary material.

---

### Decision · Program_Chairs · 2021-01-07
**Final Decision**

**Decision:**

Reject

**Comment:**

The paper proposes a Bayesian neural network model for tensor factorization, with particular focus on streaming data. The key contribution is the streaming posterior inference of the deep TF models.  The combinations of online tensor factorization, Bayesian NN with sparsity priors, posterior inference is new and interesting.  However, there are many approximation steps, and the quality of the approximation and convergence of algorithm are not well justified.